# A Novel Hybrid Internal Pipeline Leak Detection and Location System Based on Modified Real-Time Transient Modelling

Seyed Ali Mohammad Tajalli [1], Mazda Moattari [1,2,*], Seyed Vahid Naghavi [3] and Mohammad Reza Salehizadeh [1]

[1] Department of Electrical Engineering, Marvdasht Branch, Islamic Azad University, Marvdasht 7371113119, Iran; seyedalimohammad.tajalli@iau.ir (S.A.M.T.); salehizadeh2000@yahoo.com (M.R.S.)

[2] Mechateronic & Artificial Intelligence Research Center, Marvdasht Branch, Islamic Azad University, Marvdasht 7371113119, Iran

[3] Engineering Devision, Reseach Institute of Petroleum Industry, Tehran 1485733111, Iran; naghaviv@ripi.ir

* Correspondence: moattari@iau.ac.ir

**Abstract:** A This paper proposes a modified real-time transient modelling (MRTTM) framework to address the critical challenge of leak detection and localization in pipeline transmission systems. Pipelines are essential infrastructure for transporting liquids and gases, but they are susceptible to leaks, with severe environmental and economic impacts. MRTTM tackles this challenge with a three-stage operational process. First, "Data Collection" gathers sensor data from designated observation points. Second, the "Detection" stage identifies leaks. Finally, "Decision-Making" utilizes MRTTM to pinpoint the exact leak magnitude and location. This paper introduces an innovative method designed to significantly enhance pipeline leak detection and localization through the application of artificial intelligence and advanced signal processing techniques. The improved MRTTM framework integrates AI for pattern recognition, state space modelling for leak segment identification, and an extended Kalman filter (EKF) for precise leak location estimation, addressing the limitations of traditional methods. This paper showcases the application of MRTTM through a case study using the K-nearest neighbors (KNN) method on a water transmission pipeline for leak detection. KNN aids in classifying leak patterns and identifying the most likely leak location. Additionally, MRTTM incorporates the EKF, enabling real-time updates during transient events for faster leak identification. Preprocessing sensor data before comparison with the leakage pattern bank (LPB) minimizes false alarms and enhances detection reliability. Overall, the AI-powered MRTTM framework offers a powerful solution for swift and precise leak detection and localization in pipeline systems. The functionality of the framework is examined, and the results effectively approve the effectiveness of this methodology. The experimental results validate the practical utility of the MRTTM framework in real-world applications, demonstrating up to 90% detection accuracy and an F1 score of 0.92.

**Keywords:** pipeline leak detection; real-time transient modelling; extended Kalman filter; support vector machine; K-nearest neighbors

## 1. Introduction

### 1.1. Motivation

Fluid distribution systems are critical components of various industries worldwide. However, these pipelines are susceptible to leaks, which can cause environmental hazards and significant damage to infrastructure. As such, it is essential to develop an effective leak detection system (LDS). Pipeline LDSs are crucial for ensuring the safe and efficient operation of pipelines. These systems are designed to detect leaks in pipelines quickly and accurately, which is critical for preventing environmental damage, protecting public safety, and minimizing the potential for costly repairs or downtime. Standards in terms of

leak detection are published by the API (Washington, DC, USA) and TRFL (Germany) [1]. Pipeline leaks pose a significant threat to public safety, environmental well-being, and economic stability. Regulatory agencies like the Pipeline and Hazardous Materials Safety Administration (PHMSA) [2] enforce strict leak detection standards for pipelines [3]. However, existing leak detection methods often face limitations, particularly when dealing with aging infrastructure or complex operating conditions [4,5]. This research is motivated by the need for more robust and adaptable leak detection systems that can address these limitations. The current focus on hardware-based approaches can be restrictive in situations where infrastructure upgrades are impractical. This paper proposes a software-centric solution that leverages the power of AI to enhance leak detection performance.

### 1.2. Literature Review

Various leak detection methods have been employed in recent years to keep track of a pipeline's integrity [6–8]. In accordance with the physical principles that influence LDS, leak detection methods can be divided into different categories. As illustrated in Figure 1, three categories of leak methods for detection are present: exterior/hardware methods, interior/computational software methods, and visual/biological methods. Exterior methods rely on sensors along the pipeline to detect changes in external parameters like pressure and temperature, while interior methods use technical instrumentation outputs to track internal pipeline variables and analyze algorithmic analysis instruments. Visual/Biological methods involve physically inspecting the pipeline or using biological indicators to detect leaks. The choice of method depends on the pipeline operator's specific needs, as each type of method has its advantages and limitations. Exterior methods are generally less expensive and easy to install but less sensitive to small leaks. Interior methods are highly accurate but require specialized equipment and expertise. Visual/biological methods are typically used with other methods and can provide an additional layer of safety and protection. The focus of this paper is on internal methods, and selected methods are further analyzed and reviewed. Meanwhile, internal systems use technical instrumentation outputs that track the internal pipeline variables, and algorithmic analysis instruments are also regarded as computational pipeline monitoring (CPM) systems [9].

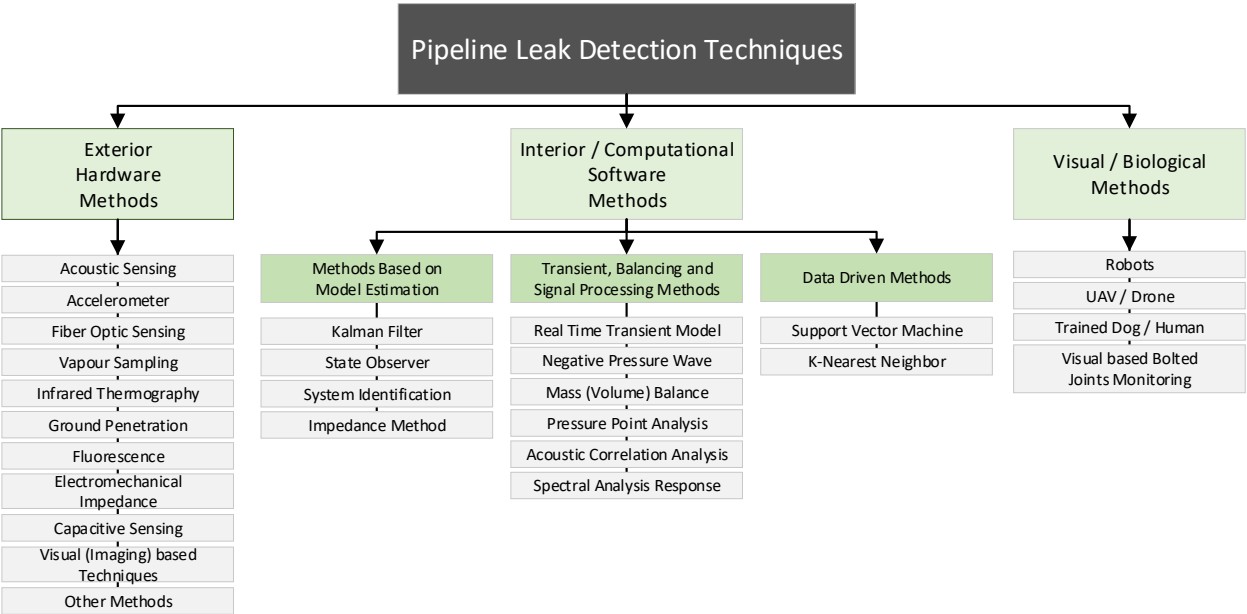

**Figure 1.** Classification of leak detection techniques [10–12].

Pipeline LDS uses mathematical models of the pipeline, along with measurements of flow rates, pressures, temperatures, and additional variables in order to determine the pipeline's state (i.e., the parameters that describe the physical condition of the pipeline) at

any given time. Pipeline LDSs using state estimators or observers are designed to monitor pipelines and detect leaks immediately as they occur. The state estimators or observers have been widely used for fault diagnosis of pipelines and estimating state variables in real time. Fault diagnosis [7] is a practical technique based on observation, which includes detecting, isolating, and identifying system failures. The observation system is essential in deterministic model-based fault diagnostic approaches given that it maintains updating determined models [13]. The analysis of residues representing faults is carried out in an observer-based analytical approach by estimating process efficiency and using residual estimation errors [14]. Different state estimators or observers, such as EKF [15], unscented Kalman filter (UKF) [16], particle filter (Monte Carlo filter) [17], recursive least squares (RLS) [18], moving horizon estimation (MHE) [9], Luenberger observer [19], and event-triggered particle filter [20] can be utilized in pipeline LDS. These filters use a set of equations to estimate the current state of the pipeline based on past measurements and the current control inputs [21].

However, based on the previously mentioned requirements, there is an operational method known as RTTM. Using mathematical algorithms, RTTM can instantly determine the mass flow, pressure, density, and temperature at any location throughout the pipeline. In order to create the local profiles, which comprise the pressure profile and transient behavior of the pipeline, temperature and pressure sensors are placed at the head stations [22]. RTMM has several advantages over other pipeline leak detection methods. First, it is capable of detecting leaks of any size, including small leaks that may go undetected by other methods. Second, it is not affected by changes in the pipeline's operating conditions, making it more reliable than other methods. Third, it can be used to identify the location of a leak, allowing for more targeted repairs and minimizing downtime. However, there are also some drawbacks to the RTTM method. One of the main limitations is the computational resources required to run the hydraulic models in real-time. This can be a significant challenge, especially for large pipelines or those with complex geometries. Additionally, RTTM requires an exceptional amount of capacity to develop and maintain the hydraulic models. On the other hand, one of the most significant methods in the field of leak detection is the E-RTTM approach, which was created by the KROHNE company (PipePatrol, Duisburg, Germany) and has been successfully utilized in this sector [23]. E-RTMM is an advanced pipeline leak detection method that builds upon the principles of RTMM by incorporating additional data sources and ML techniques to improve detection accuracy and reduce false alarms. Furthermore, the industry is much improved by AI, and a variety of ML-based AI algorithms have been applied to find anomalies in pipelines [24]. In a prior study [25], for the purpose of identifying pipeline leaks, many ML models were put into use. The one that follows analyzes different methods used in ML like linear regression, decision trees, support vector machines (SVMs), naive Bayes, and KNN to find pipeline leaks using simulated data from industrial processes. Following is a more detailed discussion of the investigation and evaluation of classifier training methods, along with efforts to increase the accuracy of leak location identification. The SVM is a statistical learning principal learning algorithm. The SVM has become the most widely had been using learning machine for data classification and regression that utilizes a model of supervised learning. The representation of the optimal hyperplane that functions as a boundary between the two classes is the main objective of SVM in data classification [26]. SVMs used to be successfully applied to a variety of high-dimensional and nonlinear learning issues. Meanwhile, for classification and regression, KNN [27] is a supervised ML method. It is a non-parametric approach that does not rely on any underlying assumptions about the distribution of the data, even though it only contains data, even during the training process, without having to perform any mathematical calculations upon that. This algorithm generates a model-based framework that anticipates the proper class for the testing dataset by calculating the distance between the testing data and the training data. The k-points closest to the testing dataset are determined using the algorithm. The classification with the highest probability is then chosen after computing the probability that the test data will be categorized into

each participating class. The parameter k indicates the number of neighbors' relatives who might be allowed to vote. The difference between a point and its nearest neighbor can be calculated using a variety of methods, which is mentioned in [28].

### 1.3. Research Gaps

The research gap in this field evidences a need to develop more accurate and reliable methods for pipeline leak detection that can overcome the limitations of ERTTM. While ERTTM is a widely used method, it still has several limitations that need to be addressed. One of the issues with ERTTM is that it relies on precise modelling of the pipeline and its surrounding environment. Even small variations in the pipeline's geometry, soil properties, or temperature can significantly affect the model's accuracy. To address this problem, researchers are exploring the use of advanced modelling techniques, such as ML algorithms, that can adapt to changes in the pipeline's operating conditions. Another problem is the presence of noise in the measurement data. This can arise from various sources, such as measurement errors, sensor drift, or interference from nearby equipment. Researchers are looking into using sophisticated signal processing methods to solve this issue, such as wavelet analysis or Kalman filtering, which can reduce unwanted noise in the data and increase the accuracy of the LDS. From another point of view, a successful state reconstruction requires certain inputs to obtain a sufficient outcome, and one practical technique for developing real-time transient a reconstruction would be to construct an EKF.

### 1.4. Methodology

To address the research gaps identified earlier, this paper presents an effective method for identifying leaks in pipelines when the size and location of the leaks are unknown. The proposed method involves modelling the pipeline and simulating multiple dynamic states using simulation software to obtain more precise estimates for leak detection and localization. A comprehensive framework capable of analyzing and performing multiple processes in parallel is essential for achieving an optimal solution.

The MRTTM framework is introduced as a solution that integrates these methods to offer an efficient approach for identifying leaks in pipelines with unknown sizes and locations. This framework is structured to enable parallel processing through a modular design, where each module is responsible for specific tasks such as data preprocessing, pattern recognition, state space modelling, and leak localization. The parallel execution of these processes is coordinated by a control system that ensures synchronization and seamless data flow between modules, thereby optimizing the performance and accuracy of the MRTTM framework in real-time applications. Precision, recall, specificity, and F-score are among the metrics employed to evaluate the performance of this framework, alongside accuracy.

### 1.5. Contributions

This work proposes a novel framework MRTTM that leverages AI for pipeline leak detection and location. The MRTTM framework offers several key contributions:

Development of a leak detection and location system: The framework utilizes an LPB that stores and adapts to pipeline data under various operating conditions.

AI-powered leak detection with adaptability: Machine learning algorithms are employed to analyze data from the LPB and the RTTM method. The most accurate model is selected for real-time leak detection, ensuring adaptability to changing conditions.

Comparative Analysis of machine learning algorithms: Two ML algorithms are compared for both leak detection and LPB creation, offering valuable insights into their effectiveness.

Comprehensive evaluation methodology: The framework's performance is evaluated using metrics like F1 score, precision, recall, accuracy, and specificity, providing a robust assessment.

Enhanced leak location accuracy: The EKF is integrated into the framework to improve leak location estimation by filtering out background noise and flow-pressure signal changes.

This significantly increases the accuracy of the leak detection and location algorithm operating in a real-time environment.

### 1.6. Paper Organization

The remaining sections of this paper are arranged as follows: Section 2 includes a description of the pipeline model as well as an assessment of problems for which solutions have been investigated. Section 3 introduces the concept of the MRTTM method and proposed methodology using a real-time framework leveraging AI for enhanced pipeline leak detection and localization. Section 4 attempts to incorporate numerical simulations into the experimentation and analysis to demonstrate the features of MRTTM and looks into the final results of the offered model. Section 5 concludes this study by offering this paper's findings and conclusions.

## 2. Problem Formulation

There is always the possibility that a pipeline will leak. One of the hardest challenges in this field is precisely locating and measuring leaks. In this paper, an additional issue that we attempt to explore is a suggested modelling method for precisely imitating the performance of a pipeline with leakage. The following is a description of the fluid dynamics consistency and dynamic equations in a pipeline [29]: As seen in Figure 2, the complete pipeline model along with the pump is placed [30], which is specified in using a dynamic real-time transient model of pipeline values $H$ (pressure) and $Q$ (flow rate). The model that is suggested separates the pipeline into n segments, with $Q_1$-$Q_n$ and $H_1$-$H_n$ representing the flow and pressure at the beginning and end of each segment. The head pressure at the beginning $H_{in}$ and the head pressure at the end $H_{out}$ are also displayed. The measured difference between real and simulated input and output are $R_{in}$ and $R_{out}$. Thermal convection velocity variations, fluid density, multi-dimensionality, and multi-pipe area are all held constant in the proposed model.

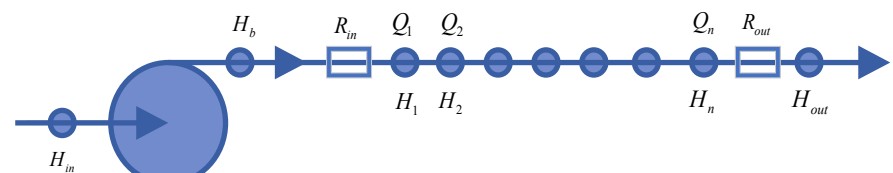

**Figure 2.** The complete model of the pipe with a pump.

The environment's wave velocity is significantly higher than the rate of fluid flow notwithstanding these considerations. Additionally, the pipe's length is adequate to keep the pattern of lateral flow constant. This flow pattern is defined by a pair of nonlinear hyperbolic partial differential equations (PDEs) that arise from the application of mass and momentum constraints across a predefined control length distance [31].

In order to create a set of initial PDEs formed by momentum Equation (1) and continuity Equation (2), the mass and momentum conservation laws are applied to a control volume that takes consideration of a pipeline.

$$\frac{\partial q(z,t)}{\partial t} + gA\frac{\partial h(z,t)}{\partial z} + \frac{f(z,t)}{2Ad}q(z,t)|q(z,t)| = 0 \tag{1}$$

$$\frac{\partial h(z,t)}{\partial t} + \frac{c^2}{gA}\frac{\partial q(z,t)}{\partial z} = 0 \tag{2}$$

where $h$ is the pipeline's pressure head (m), $q$ is the pipeline's flow rate (m$^3$/s), $c$ is the fluid's wave speed (m/s), $g$ is the pipeline's gravitational acceleration (m/s$^2$), $A$ is the pipe's cross-sectional area (m$^2$), $d$ is the pipe's diameter (m), $f$ is the coefficient of friction, and $t$ and $z$, respectively, are the time (s) and space (m) coordinates. In addition, the general

equation that defines the conduct of leakage in the pipeline is derived from the Bernoulli equation.

$$q_L = \lambda \sqrt{h_L} \tag{3}$$

where $q_L$ is the leakage stream, $h_L$ is the leakage pressure head, and parameter $\lambda$ is the leakage constant, which depends on the region of the leakage.

$$q_1 = q_2 + q_L \tag{4}$$

As seen in Equation (4), the sum of the leakage flow rate $q_L$ and the output flow rate $q_2$ from the pipe after the leakage equals the input flow rate $q_1$ at the beginning of the pipe [32].

Several prototype failure and leakage detection approaches regard coefficient $f$ to be consistent; however, it is frequently adjusted whenever a leak is found. Moreover, the coefficient is affected mostly by the Reynolds number. In combination with the Darcy–Weisbach formula, pipe wall friction is found to be an excellent contributor to pipe pressure drop [33]. As well as precision in the calculation in accordance with the elements in the Moody chart, the Darcy–Weisbach friction factor against Reynolds number (*Re*) for different amounts of relative roughness is a crucial factor in recognizing and diagnosing leakage. For each observer iteration, the friction factor $f$ is modified for both the $q_1$ and $q_2$ flow rates to compensate for flow system modifications due to leakage. Since it requires an inner iteration to calculate $f$ from the implicit Colebrook–White formula [34], with extra mathematical effort, Swamee–Jain [35] clear approximation could be used instead. The friction factor $f$ can be calculated as follows:

$$f(q) = 0.25 \left( log_{10} \left( \frac{\varepsilon}{3.7} + 5.74 \left( \frac{Av}{qd} \right) \right) \right)^{-2} \tag{5}$$

The ODE system reflects the dynamic model of the pipeline:

$$\dot{q}_1 = \frac{gA}{z_l}(h_1 - h_2) - \frac{f(q_1)}{2Ad} q_1 |q_1| \tag{6}$$

$$\dot{h}_2 = \frac{c^2}{gAz_l} \left( q_1 - q_2 - \lambda \sqrt{|h_2|} \right) \tag{7}$$

$$\dot{q}_2 = \frac{gA}{L - z_l}(h_2 - h_3) - \frac{f(q_2)}{2Ad} q_2 |q_2| \tag{8}$$

Hydraulic transient models are constructed in accordance with the rules governing transient mass flow and dynamic preservation [36]. To attain precise leak detection and localization in pipelines, it is imperative to employ model parameters that align with the observed pipeline characteristics and utilize advanced techniques for leak identification. ML models present a promising solution capable of accurately detecting leaks while minimizing false positives, operating in real-time to furnish timely information regarding leak locations. The integration of EKF into the leak detection process significantly enhances the accuracy of location estimation. Regular validation of simulated leak estimates through comparison with actual observed leaks is paramount.

RTTM, involving the analysis of pressure and flow measurements in the pipeline, proves to be an effective method for leak detection. When combined with ML models and EKF, RTTM further enhances accuracy in leak localization. The comprehensive discussion of the methodologies mentioned earlier is expounded upon in the subsequent section.

## 3. Leak Detection and Accurate Leak Location

In this section, the author deliberates upon the proposed methodologies in connection with the identified research gap. The initial segment delved into the overarching aspects of the proposed plan, emphasizing the integration of the process within the framework

of real-time operations. Subsequently, the second section expounded upon the intricacies and functioning of the MRTTM method. Collectively, the envisaged methods are geared towards augmenting the efficiency and effectiveness of the MRTTM approach through seamless integration into a real-time framework empowered by AI.

*3.1. AI-Empowered MRTTM Framework*

Based on the outlined diagnostic procedures and the identified challenges, the proposed solution encompasses a series of systematic steps for the detection and localization of leaks in pipelines. Initial detection involves diagnostic testing, utilizing both the physical and dynamic characteristics of the pipelines. Subsequently, the magnitude of the detected leakage is assessed through the implementation of the RTTM method and the analysis of data collected by the observer. Following this, pipeline segments are analyzed using simulation software, and diverse leakage scenarios, encompassing large, medium, and small leakages, are scrutinized for each segment. The outcomes of various tests under anticipated conditions are systematically classified into distinct categories. The selection is then juxtaposed with the optimal training algorithm, culminating in the formulation of the LPB.

In the process of developing the MRTTM framework, multiple machine learning algorithms were evaluated to identify the most suitable model for pipeline leak detection. Among these, the KNN model was selected due to its simplicity, adaptability, and robustness in handling noisy data, as well as its stable performance under varying conditions. The KNN model consistently outperformed other models such as SVM and decision trees in this specific context. While SVM demonstrated high accuracy in controlled environments, it struggled with noise and required substantial computational resources, limiting its real-time applicability. Decision trees, despite their interpretability, exhibited overfitting tendencies and inconsistent performance across different pipeline conditions. The KNN model's ability to generalize effectively, coupled with its resilience to noise, made it the optimal choice for the MRTTM framework, balancing accuracy, computational efficiency, and robustness for real-time leak detection. Ultimately, the MRTTM method is employed to achieve the highest accuracy in detecting the location and magnitude of the leak under varied dynamic states of the pipeline. Following leak detection and the determination of the affected pipeline segment, the trained algorithm considers two hypothetical leakages at the initiation and conclusion of the diagnostic pipe section. The actual location of the leakage is then estimated utilizing an EKF observer, which discerns all pipeline parameter rates (pressure and flow rate values) amidst process and measurement noise. Therefore, the integration of all suggested processes is imperative, and a comprehensive depiction of the proposed remedy is elucidated in Figure 3. The conceptual real-time framework comprises three integral components: the virtual pipeline (data collection), leak detection (detection), and MRTTM (decision-making). Within the virtual pipeline segment, an observer analyzes sensor data in real time to glean information regarding the dynamic state of the pipeline. The detection section employs diagnostic methods tailored to various dynamic fluid conditions for leak detection. Upon detecting a leak, the decision-making stage is initiated, utilizing the proposed MRTTM to estimate the leak's location based on the pipeline's conditions. As illustrated in the figure, the MRTTM framework leverages artificial intelligence by first transferring data from the LPB. These data are then used to train machine learning models like KNN, SVM, and others, with the model exhibiting the highest accuracy being selected after the initial training process. Subsequently, the output data from these trained models are evaluated using test data that were previously processed by the RTTM method. Finally, the EKF is employed to achieve the most accurate estimation of the leak location.

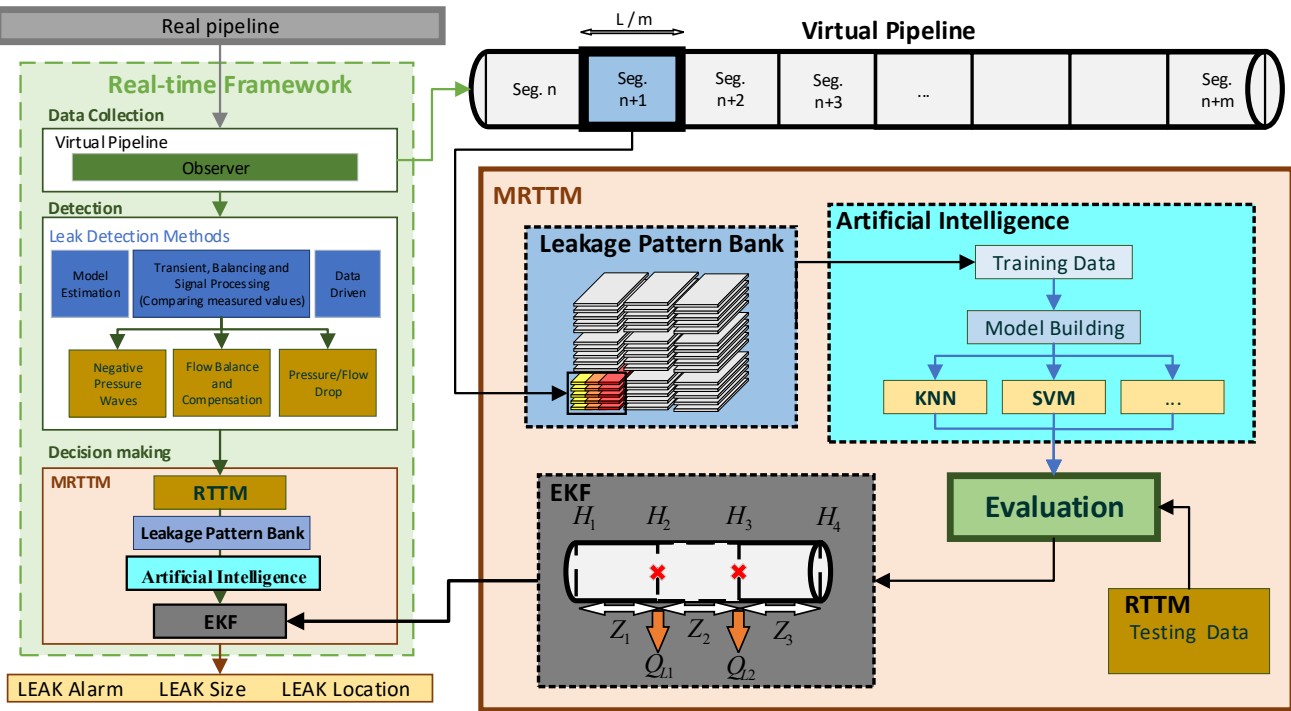

**Figure 3.** Concept of MRTTM using real-time framework empowered by AI.

Furthermore, the proposed conceptual real-time framework represents an integrated approach to enhancing pipeline safety, leveraging the synergy of horizontal and vertical techniques. The horizontal component incorporates API 1130 procedures while strictly adhering to guidance from TRFL and PHMSA. This alignment fortifies the system with industry-supported best practices, augmenting reliability and effectiveness. The sophisticated LPB and vertical fusion of AI methods and EKF for real-time updates during transient events complement this horizontal integration. In broad terms, the proposed solution furnishes a reliable and accurate method for real-time pipeline leak detection, ensuring the safe and efficient operation of pipelines. This systematic approach combines diagnostic methods, simulation software, and advanced algorithms to detect leaks and estimate their location with high precision.

### 3.2. Modified Real-Time Transient Modelling (MRTTM) Method

The key distinction between RTTM [37] and other methodologies is its method for modelling all dynamic fluid characteristic features, including flow, pressure, and temperature, in addition to the general structure of physical pipeline details including length, diameter, thickness, and product properties, such as density and viscosity. The pipeline observer is at the core of the leak monitoring system [38]. It denotes the flow fundamentals and thermodynamic parameters of the leak-free pipeline anywhere along the length. For the entire purpose, pressures $P_I$ and $P_O$ are determined at the inlet and outlet. The temperatures of the fluid and density are also necessary. These principles are used to calculate the mass flow $\dot{M}_I$ and $\dot{M}_O$. These estimated values were compared to the measured values, generating the residuals $x \equiv M_I - \dot{M}_I$ and $y \equiv M_O - \dot{M}_O$. When there is a leak, deviations take place wherein the leak location $X_{Leak}$ and the leak rate (speed, flow rates, or mass flow) $\dot{M}_{Leak}$ could be calculated [22]:

$$\dot{M}_{Leak} = x - y \tag{9}$$

$$X_{Leak} = \frac{-y}{x - y} L \tag{10}$$

However, the major idea is to build a smart real-time transient model using ML that allows for the correlation between the pipeline model input and output data as a mathematical function. Leak factors can be reconstructed by adding them in a new, enhanced state space model as an existing state parameter. The EKF is required for estimating leakage variables and filtering out background noise and flow–pressure signal change in the real-time transient model state as part of the diagnostic leakage feature. EKF is a straightforward installation approach that is especially useful for nonlinear systems with workable applications. A state space definition of fluid flow dynamics with $H_i$ and $Q_i$ as vector $x$ state variables, boundary conditions as known u-vector variables for input, and the typical model makes it easy to obtain parameters in the form of y-vector output variables [39]. As a consequence, the problem of leak detection can be solved by developing a system status observer based on the modified state matrix developed in [40]. Theoretically, the Kalman filter is a state estimator based on the statistical definition of noise at the measured output of the linear dynamic system for the optimal estimation of the undetermined state.

To accomplish this, the parameters $z_l$ and $\begin{bmatrix} y_1 \\ y_2 \end{bmatrix} = \begin{bmatrix} 1 & 0 & 0 & 0 & 0 \\ 0 & 0 & 1 & 0 & 0 \end{bmatrix} \begin{bmatrix} x_1 \\ x_2 \\ x_3 \\ x_4 \\ x_5 \end{bmatrix} + \begin{bmatrix} v_1 \\ v_2 \end{bmatrix}$ are

regarded as new state variables with dynamics, $\dot{z}_l = 0$ and $\dot{\lambda} = 0$, that could be added to the original state vector. As a result, the new augmented state vector would be as follows:

$$X = \begin{bmatrix} q_1 & h_2 & q_2 & z_l & \lambda \end{bmatrix}^T = \begin{bmatrix} x_1 & x_2 & x_3 & x_4 & x_5 \end{bmatrix}^T \tag{11}$$

As a result, the extended dynamic model is found as follows:

$$\dot{x}_1 = \frac{gA}{x_4}(u_1 - x_2) - \frac{f(x_1)}{2Ad}x_1|x_1| \tag{12a}$$

$$\dot{x}_2 = \frac{c^2}{gAx_4}\left(x_1 - x_3 - x_5\sqrt{|x_2|}\right) \tag{12b}$$

$$\dot{x}_3 = \frac{gA}{L - x_4}(x_2 - u_2) - \frac{f(x_3)}{2Ad}x_3|x_3| \tag{12c}$$

$$\dot{x}_4 = 0 \tag{12d}$$

$$\dot{x}_5 = 0 \tag{12e}$$

Implementing Heun's approach improvement to this continuous-time model ($\dot{X} = \phi(x, u)$), which has the following form, allowed for Equations (12a) to (12e) to be transformed into a discrete-time model that is specific to and appropriate for the EKF.

$$\dot{X} = x_k + \frac{T_s}{2}(\phi(x_k, u_k) + \phi(x_k + T_s\phi(x_k, u_k), u_k)) \tag{13}$$

The observation that connects the state $x$ to the measurement ($y = \begin{bmatrix} q_1 & q_2 \end{bmatrix}$) completes the requirement model to use the EKF [29]:

$$\begin{bmatrix} y_1 \\ y_2 \end{bmatrix} = \underbrace{\begin{bmatrix} 1 & 0 & 0 & 0 & 0 \\ 0 & 0 & 1 & 0 & 0 \end{bmatrix}}_{C} \begin{bmatrix} x_1 \\ x_2 \\ x_3 \\ x_4 \\ x_5 \end{bmatrix} + \begin{bmatrix} v_1 \\ v_2 \end{bmatrix} \tag{14}$$

A discrete-time EKF can be used as just another nonlinear observer to generate an observer for model Equation (14) and use the state of the extended system Equation (13). Heun's method is applied to accomplish this because it has been demonstrated to offer

a fantastic trade-off between sample selection and accuracy. Model Equation (14) is discretized in order to achieve this. The following is the alternative for the initial value issue using this method, in which $T_s$ is the time step and $k$ is the time varying index; eventually, using Equation (16), usual mathematical expressions in Equation (14) are transformed into a discrete time nonlinear model including the form:

$$X_{k+1} = \Phi(x_k, u_k, u_{k+1}) \tag{15}$$

$$Y_k = CX_K \tag{16}$$

When considering this discrete depiction (details are provided in $x$), in reality of course, all that is required would be to take into account a discrete-time EKF to others as a state observer Equation (15) without utilizing the given formula in Equation (12), which can be selected as follows:

$$K_k = P_k^- C^T (C P_k^- C^T + R)^{-1} \tag{17}$$

$$\hat{x}_k = \hat{x}_k^- + K_k (y_k - C\hat{x}_k^-) \tag{18}$$

$$P_k = (I - K_k C) P_k^- \tag{19}$$

$K$ is Kalman's gain; $\hat{x}_k$ is the expected state in time; $P_k$ is the estimation error covariance matrix.

The update equations (or prediction) of the Kalman filter time are presented by the following:

$$A_k = \left[ \frac{\delta\phi}{\delta x} \right]_{x=\hat{x}_k} \tag{20}$$

$$\hat{x}_{k+1} = \Phi(\hat{x}_k, u_k) \tag{21}$$

$$P_{k+1}^- = A_k P_k A_k^T + Q \tag{22}$$

where $\hat{x}_{k+1}$ denotes the predicted state and the $P_{k+1}^-$ describes the covariance matrix for error predicted. Kalman filter update equations (or corrections) are specified as outlined. Eventually, $R$ and $Q$ can be selected as the noise measure and processing covariance matrices. The pressure sensor, with $x^M$, will be placed next to the downstream node. Even by situation, a design of N leaks is taken into consideration for leak detection for multiple leaks [41], and the leak coordinates are $x^{L_n}$, $n = 1, \ldots, N$, $(x^{L_1} \langle \cdots \langle x^{L_N} \langle x^M \rangle$; the pipe elevation at each leak is denoted by $z^{L_n}$; $Q_0^{L_n}$ and $H_0^{L_n}$ are the steady-state discharge and head at each leak. The leak size is defined by the variable $s^{L_n} = C^d A^{L_n}$ of the lumped leak, where $C^d$ is the leak discharge coefficient and $A^{L_n}$ is the leak orifice flow area. A leak's steady-state discharge is correlated with $Q_0^{L_n} = s^{L_n} \sqrt{2g\left(H_0^{L_n} - z^{L_n}\right)}$ lumped leak parameter, wherein $g$ is the gravitational acceleration. The amounts at $x^M$ can be calculated in a corresponding way, considering the discharge $q(x^U)$ and head $h(x^U)$ equations found in [42].

The Kalman filter was utilized in this paper to detect leaks more accurately due to its advantages. As a result of the RTTM method's detection and the trained algorithm's detection of which section of the pipeline is located, two fictitious leaks are created at known locations $X_{L1}$ and $X_{L2}$ (the location of the beginning and ending of the identified pipeline segment) shown in Figure 4. Assuming steady-state conditions, the temporal terms within Equations (1) and (2) disappear and the equations become as follows:

$$\frac{\partial h}{\partial z} = \frac{fq|q|}{2gdA^2} \tag{23}$$

$$\frac{\partial q}{\partial z} = 0 \tag{24}$$

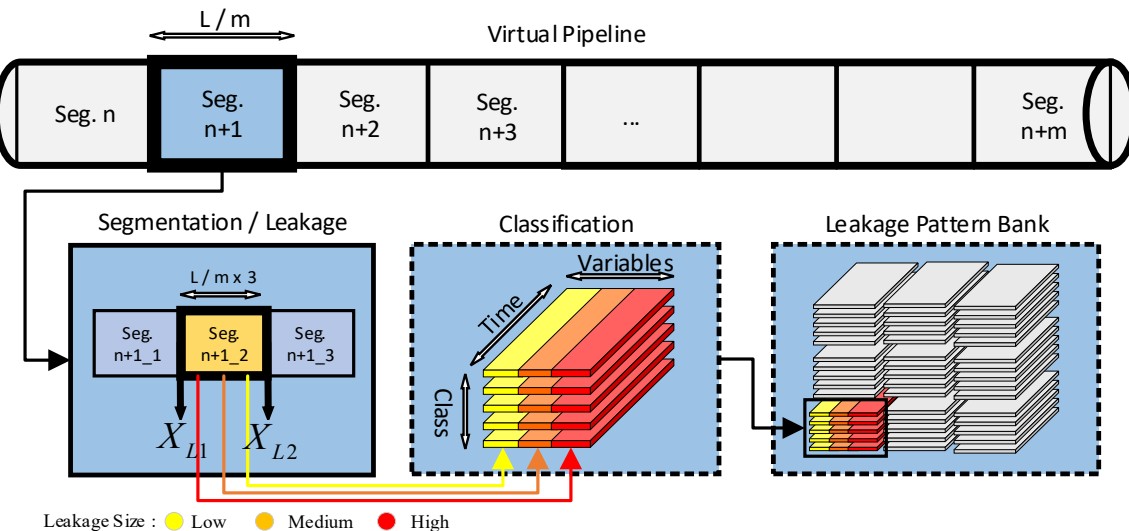

**Figure 4.** Pipeline segmentation, leakage classification, and LPB.

Equation (24) indicates that the flow rate is constant at steady state. The magnitude of the output flow rate from leakage, as a result, represents the total of the output flow rates of hypothetical leaks:

$$q_L = q_{L1} + q_{L2} \qquad (25)$$

The magnitude of the leakage can consequently be estimated using Equation (25), as well as its location, provided by the following:

$$z_L \approx \frac{q_{L1}z_{L1} + q_{L2}z_{L2}}{q_L} \qquad (26)$$

As a result, $z_L$ could be easily calculated because $z_{L1}$ and $z_{L2}$ are defined, $q_{L1}$ and $q_{L2}$ are estimated, and $q_L$ is the total magnitude of the leakages.

After the EKF calculates and estimates the quantity of leakage flow rate and the location of the leakage, the results are compared to the RTTM. This comparative analysis serves a dual purpose: enhancing leak detection accuracy and validating the efficacy of the RTTM approach. In cases where discrepancies arise, ML techniques are employed to bolster the certainty of correctness. To achieve this objective, simulation software is utilized to explore various leakage scenarios under diverse conditions. The quantity and types of these leakage scenarios are elucidated below. As illustrated in Figure 4, these requirements are implemented by initially segmenting the pipeline into several fictitious segments, each hosting a simulated leak independently. Subsequently, three distinct variations of low, medium, and high leakage are generated for each simulated leak. All scenarios run under identical conditions account for inlet and outlet pressure, inlet and outlet flow, and mass flow rate from the leakage. The LPB serves as a comprehensive database housing hypothetical leak scenarios for a given pipeline. This repository encapsulates the various simulated scenarios, facilitating a robust framework for testing and validating the proposed methodologies.

The acquired data were trained with several classifier methods in order to select the best and most ideal method for this issue. In essence, the operational procedure of the MRTTM involves the initiation of leak detection, followed by employing RTTM to narrow down the leak's location. Additionally, a recursive method known as the EKF is utilized to estimate the state of the system based on a sequence of measurements. This EKF process is integral to identifying both the location and extent of a leak when employing MRTTM for pipeline leak detection.

The EKF operates through three distinctive phases. The initialization phase is the first, wherein the EKF is configured with initial values for critical pipeline parameters like diameter, length, and roughness. During the monitoring phase, the second phase,

the EKF continually receives data from sensors strategically positioned throughout the pipeline. These data encompass fluid flow rate, pressure, and temperature. Leveraging these incoming data, the EKF calculates an estimation of the pipeline's state, encompassing the location and size of any leaks. The third phase, the estimation phase, involves utilizing the computed state of the pipeline to determine the leak's location. This is achieved by comparing the estimated output flow rate with the actual output flow rate, and the disparity between the two is leveraged to calculate both the location and size of the leak. An intrinsic advantage of the EKF lies in its remarkable precision in estimating the location of leaks. This capability is facilitated through the application of a technique known as the LPB. The LPB encompasses hypothetical scenarios that detail the precise location and flow rate of leaks both prior to and subsequent to their occurrence. Through a comparative analysis of the estimated output flow rate with these hypothetical scenarios stored in the LPB, the EKF adeptly computes the precise location of the leak. The elucidated methodology is comprehensively implemented in the subsequent section.

The sensitivity of the MRTTM framework to assumptions made in pipeline modelling, particularly concerning the consistency of the friction coefficient and varying environmental conditions, is a critical aspect of ensuring its accuracy and reliability. A sensitivity analysis was conducted to evaluate how changes in these parameters affect the framework's performance. The analysis revealed that while the MRTTM framework demonstrates robustness to minor variations in the friction coefficient, significant deviations can degrade leak detection accuracy, especially in pipeline sections with abrupt changes in material or internal surface conditions. To mitigate this, periodic recalibration of the model using real-time data is recommended to maintain optimal accuracy. Additionally, varying environmental conditions, such as changes in temperature and pressure, also influence the model's performance. The MRTTM framework adapts well to gradual environmental variations but may require further calibration in extreme conditions.

The MRTTM framework has been designed to be versatile and adaptable, making it suitable for a wide range of pipeline conditions, including oil, gas, and water transport pipelines. Its adaptability is evident in its ability to handle diverse operational environments and fluid types, effectively managing variations in pipeline configurations and operational conditions through the integration of advanced signal processing and AI-driven pattern recognition. This ensures the framework's effectiveness and reliability across different pipeline settings, enhancing its utility and robustness in real-world applications. Moreover, the MRTTM framework is compatible with existing pipeline monitoring systems, enabling real-time leak detection through its modular design. The integration of digital twin technology [43] within the MRTTM framework further enhances its capabilities by creating a virtual replica of the physical pipeline system. This allows for more accurate simulations and predictive analytics, improving leak detection and overall system reliability. However, integrating the MRTTM framework and digital twin technology into existing systems may present certain challenges. These include ensuring data compatibility, as existing systems might use different data formats and protocols, requiring the development of interfaces or adapters for seamless data exchange. Additionally, the real-time implementation of the MRTTM framework and the continuous updating of the digital twin demand sufficient computational resources, particularly for processing large datasets and running the EKF, which may necessitate hardware upgrades or the use of cloud-based solutions. System integration also poses potential difficulties due to differences in software architectures and the need for synchronization, requiring careful planning and testing to avoid disruption of ongoing monitoring activities. Lastly, maintaining the scalability and robustness of the MRTTM framework, including its digital twin component, as pipeline networks expand is crucial, necessitating regular updates and ongoing collaboration with monitoring system providers.

## 4. Experiment and Analysis

### 4.1. Methodology and Simulation Setup

In this section, the operational details of the proposed MRTTM approach are elucidated, addressing the challenges associated with leak location and detection. The primary challenge involves determining the precise location and magnitude of the leak using the RTTM approach. The most effective classifier in each scenario is identified and explained, employing a segmented pipeline model from the LPB with boundary conditions for the EKF method. This method is utilized to scrutinize the pipeline properties, incorporating two hypothetical leaks. The most accurate estimate for the location and size of the leak is subsequently determined in Figure 5 while the resulting estimations are compared.

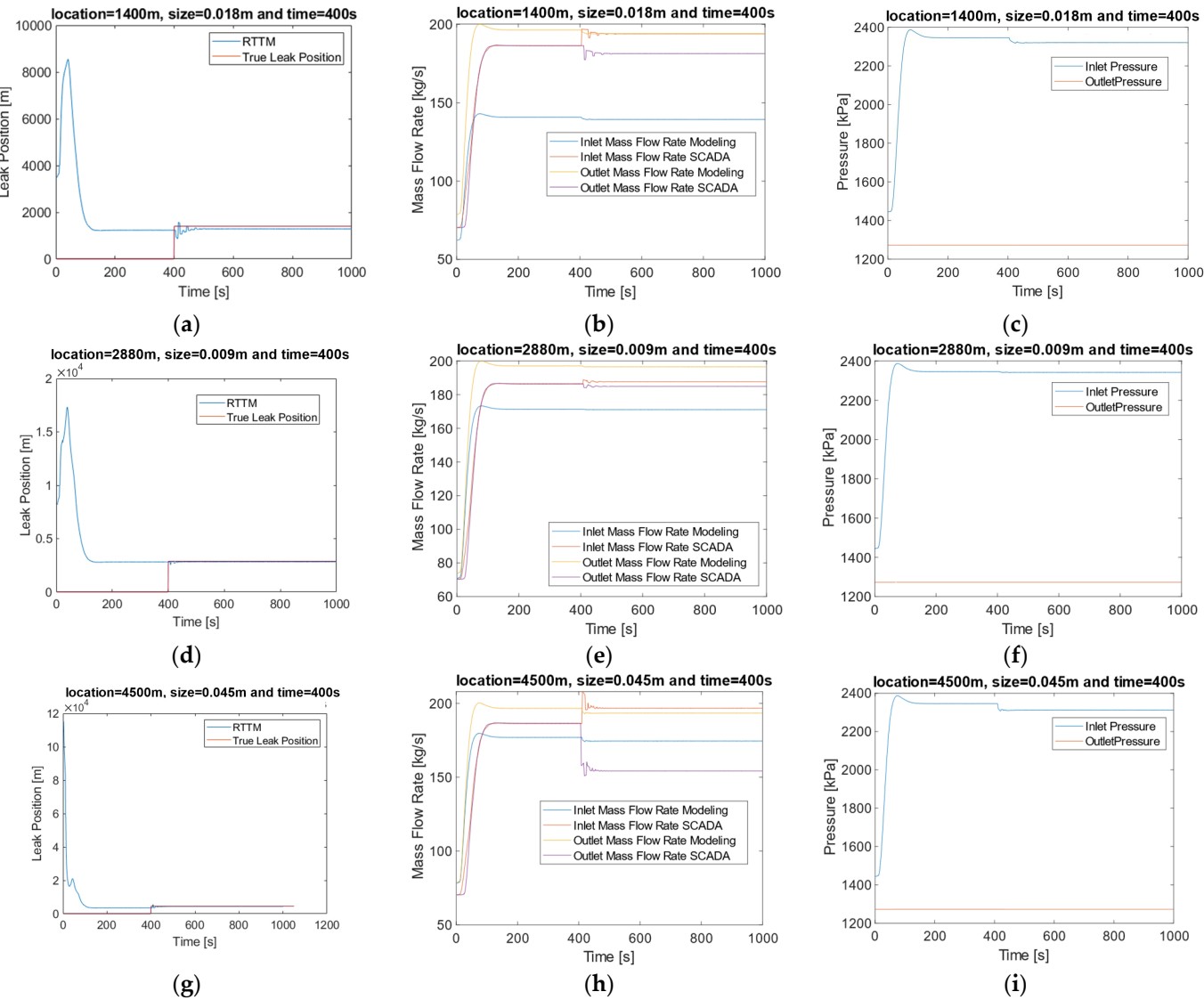

**Figure 5.** The RTTM method's characteristics for compared true leak location and RTTM method estimation (**a**,**d**,**g**), compared modelling and SCADA mass flow rate (**b**,**e**,**h**) and compared inlet and out pressure (**c**,**f**,**i**) for three test leak location.

### 4.2. Accuracy Enhancement Techniques

To enhance the accuracy of leakage estimates along the entire pipeline, the extended boundary approach technique expands the interior nodes. The application of this technique necessitates four pressure measurements to ensure the feasibility of the EKF techniques.

In the case, two internal pressure measurements, encompassing the pressures at the two boundaries, were deemed essential for the implementation of the approach.

$H_2$, $H_3$ and $Q_2$, $Q_3$ samples have been taken from the boundaries of the plant in this technique and $H_1$ and $H_4$ are estimated to be an actual pipeline. This section considers simulation results for the case studies. Length of the pipe is 6816 m, $D = 0.3$ m is the diameter, and two reservoirs and their heads are $H_1 = 150$ m and $H_2 = 130$ m. The pipe's Darcy–Weisbach friction factor is $f_{DW} = 0.012$, and the wave speed is presumed to be a = 1200 m/s. Three leaks with various sizes and locations are taken into consideration, and their details are as follows: the specification of the first test leak location = 1400 m and leak size = 6%, second test leak location = 2880 m and leak size = 3%, and third leak location =4500 m and leak size = 15%. Additionally, Figure 5 shows the characteristics of the RTTM approach for estimating mass flow, pressure, and leakage when the leak happens after 400 s.

*4.3. Classifier Performance and Scenario Analysis*

The approaches for developing an LPB are studied in this section. The pipeline is split into 24 equal sections in the first stage. There are three leaks for each segment, with values of 3%, 6%, and 15% in regard to the transmission line's diameter. Constant inlet pressure, flow rate, and leakage are generated in all models that the Simcenter Flomaster 2020.2 software has simulated in 400 s and continues simulated results for all scenarios. The current classifiers are trained by utilizing the simulated data and assigning them to the classes established in each segment of the pipeline. There is a need to train classifiers after obtaining and categorizing the data, which comprise inlet pressure, outlet pressure, inlet mass flow rate, outlet mass flow rate, and leakage mass flow rate. In order to identify the most effective method and training algorithm, different methods were tested with MATLAB R2022b software in this paper. In Table 1, the results of this comparison are presented.

**Table 1.** Comparison of SVM and KNN classifiers.

| Classifiers | Accuracy (Validation) % | Total Cost (Validation) | Prediction Speed obs/s | Training Time s |
|---|---|---|---|---|
| Fine KNN | 99.9 | 10 | 40,000 | 8.6309 |
| Weighted KNN | 99.8 | 16 | 38,000 | 10.586 |
| SVM kernel | 99.2 | 78 | 590 | 293.11 |
| Medium KNN | 99.1 | 88 | 30,000 | 11.598 |
| Cubic KNN | 99.0 | 105 | 16,000 | 9.4564 |
| Logistic regression kernel | 98.4 | 167 | 490 | 233.76 |
| Coarse KNN | 98.3 | 171 | 17,000 | 10.928 |
| Quadratic SVM | 98.1 | 195 | 2800 | 401.28 |
| Linear SVM | 95.9 | 424 | 2300 | 102.53 |
| Cosine KNN | 94.7 | 553 | 4400 | 13.619 |
| Fine Gaussian SVM | 88.4 | 1205 | 1300 | 455.64 |
| Medium Gaussian SVM | 63.2 | 3805 | 750 | 541.64 |
| Cubic SVM | 58.9 | 4251 | 4500 | 1219.6 |
| Coarse Gaussian SVM | 38.3 | 6390 | 550 | 661.47 |

It is possible that, from an operational and practical standpoint, in some circumstances, the optimal measuring sensors cannot provide four input variables for training the algorithm. In this instance, predefined scenarios are taken into consideration.

The first scenario, in which just pressure and flow rate are available at the pipeline's inlet and outlet (four variables).

The second scenario, which just has input and output flow rate (two variables).

The third scenario, which just has input and output pressure (two variables).

Table 2 compares the current scenarios with three of the best high-accuracy classifiers.

The results of the studies demonstrate that the KNN classifier is more accurate than other methods. As a consequence, this paper's testing and evaluation were conducted using the KNN classifier. The characteristics of three leakage locations that were discussed in the previous section are used in the following to evaluate the trained algorithm.

**Table 2.** Compared three scenarios with the best high-accuracy classifiers.

| Scenario | Classifier | Accuracy (Validation) % | Total Cost (Validation) | Prediction Speed obs/s | Training Time s |
|---|---|---|---|---|---|
| | Fine KNN | 99.8 | 21 | 50,000 | 3.5869 |
| Scenario 1 | Weighted KNN | 99.8 | 25 | 41,000 | 6.861 |
| | SVM Kernel | 97.4 | 267 | 1300 | 161.08 |
| | Fine KNN | 98.3 | 173 | 62,000 | 2.2025 |
| Scenario 2 | Weighted KNN | 98.4 | 164 | 51,000 | 5.2388 |
| | SVM Kernel | 94 | 625 | 1600 | 136.06 |
| | Fine KNN | 98.3 | 173 | 80,000 | 2.7874 |
| Scenario 3 | Weighted KNN | 98.3 | 175 | 64,000 | 6.9076 |
| | SVM Kernel | 35.7 | 6653 | 2400 | 88.053 |

### 4.4. Detailed Results and Comparison

Tables 3 and 4 in this section contain detail performance data for the proposed method with different pipeline section numbers. It is discussed whether adding pipeline sections impacts accuracy. Furthermore, the contribution of the Kalman filter to improving detection accuracy is investigated. The information in Table 3 indicates that the estimated accuracy decreases with a reduced sensor number; however, it should be noted that there may not be a pressure or flow measurement sensor in real conditions or that the sensor may not have the correct measurement precision. Ideally, scenarios 2 and 3 can be thought of as a backup to be used in unusual circumstances. On the other hand, increasing the number of tested sections can improve decision-making accuracy and robustness as well as machine learning's potential to train the algorithm. For instance, there were 24 pipeline sections under the previous circumstances. Each segment is split into three segments in order to optimize estimation, and three different leakage modes—large, medium, and small—are taken into consideration for each segment.

**Table 3.** Scenario description and results of applying the proposed method for 24 pipeline sections (24 classes).

| Scenario Description | Scenario 1 | Scenario 2 | Scenario 3 |
|---|---|---|---|
| True positive | 32 | 22.6 | 14.42 |
| False positive | 5.5 | 7.4 | 7 |
| False negative | 5.5 | 7.4 | 7 |
| True negative | 107 | 112.6 | 121.57 |
| Precision | 85% | 75% | 67% |
| Recall | 82% | 79% | 62% |
| Specificity | 95% | 93% | 94% |
| Accuracy | 85% | 75% | 67% |
| F1 score | 88% | 79% | 71% |

**Table 4.** Scenario description and results of applying the proposed method for 24 pipeline sections (72 classes).

| Scenario Description | Scenario 1 | Scenario 2 | Scenario 3 |
|---|---|---|---|
| True positive | 22 | 23.2 | 16 |
| False positive | 3 | 6.8 | 5 |
| False negative | 3 | 6.8 | 5 |
| True negative | 122 | 113.2 | 121.32 |
| Precision | 84% | 76% | 68% |
| Recall | 82% | 80% | 63% |
| Specificity | 97% | 94% | 93% |
| Accuracy | 88% | 77% | 69% |
| F1 score | 91% | 82% | 72% |

The data presented in Table 4 clearly demonstrate that an increase in the number of pipeline sections corresponds to an improvement in leak detection accuracy. The perfor-

mance of the trained algorithm exhibits enhanced accuracy with an augmented number of measurement tests conducted on the pipeline. Consequently, the LPB acquires a more dependable leakage pattern. In this context, the Kalman filter is employed for detection, providing a higher estimate. As elucidated earlier, the Kalman filter is applied to achieve more precise detection and subsequent comparison. After the RTTM method estimates the location and magnitude, and the leak pattern bank determines the respective class, the Kalman filter contributes to a refined and more accurate detection process. Figure 6 illustrates the layout of two hypothetical locations of fictitious leakage location at the beginning and end of the leakage range determined by the classifier diagnostic method. The simulation output data contains the inlet pressures and outlet pressures in each pipeline segment ($H_1$, $H_2$, $H_3$, and $H_4$), flow and outlet flow in each pipeline segment ($Q_{11}$, $Q_{12}$, $Q_{21}$, $Q_{22}$, $Q_{31}$, and $Q_{32}$), and the outlet flow from two fictitious leaks ($Q_{L1}$ and $Q_{L2}$).

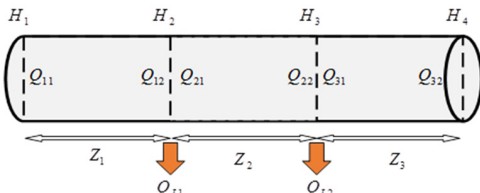

**Figure 6.** Hydraulic pressure indicators and ow rates with leakage presentation along the transmission line.

Figure 7 shows three pipelines with identical specifications and fictitious leaks constructed with Simcenter Flomaster 2020.2 software. The obtained leak is a real leak between two unrealistic leaks. As discussed in previous sections, the more accurate location of the leak is estimated in this approach using the EKF.

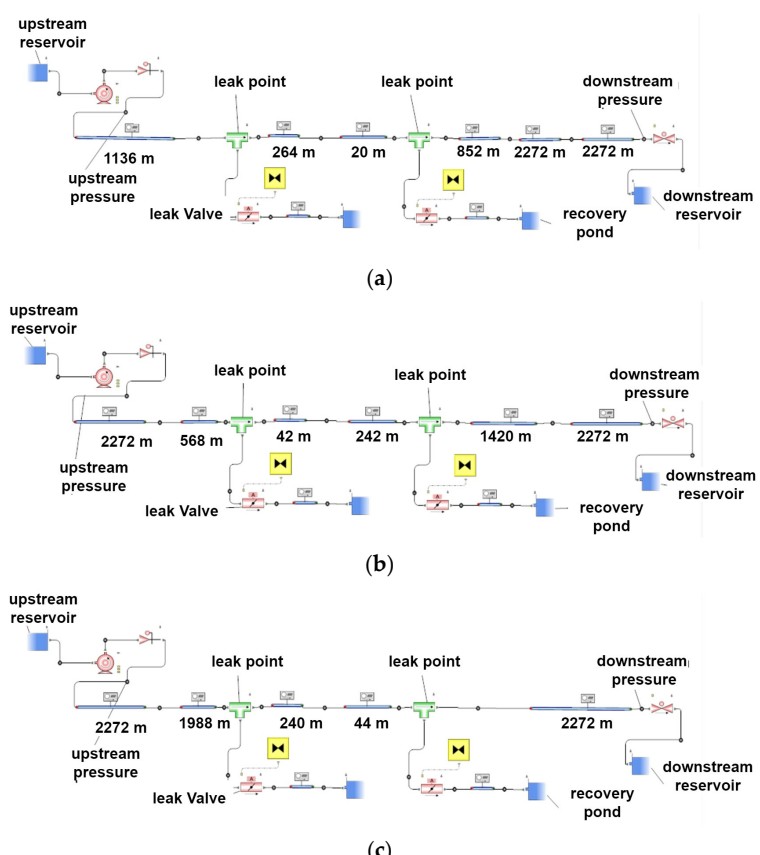

**Figure 7.** Pipeline simulation design with two fictitious leaks. (**a**) for true leakage at 1400 m; (**b**) for true leakage at 2880 m; (**c**) for true leakage at 4500 m.

The flow and pressure diagrams are represented in Figures 8–10. The estimated model using the Kalman method along with RTTM and the outcome of the observed training classifier are included in the diagrams, as is the simulation using Simcenter Flomaster 2020.2. Data from simulation and modelling methods were used to calculate different locations and the magnitude of three leaks, which are covered in more detail in the preceding section. On the other hand, the data represented by EKF were able to deliver an improved result by lowering the unwanted noise.

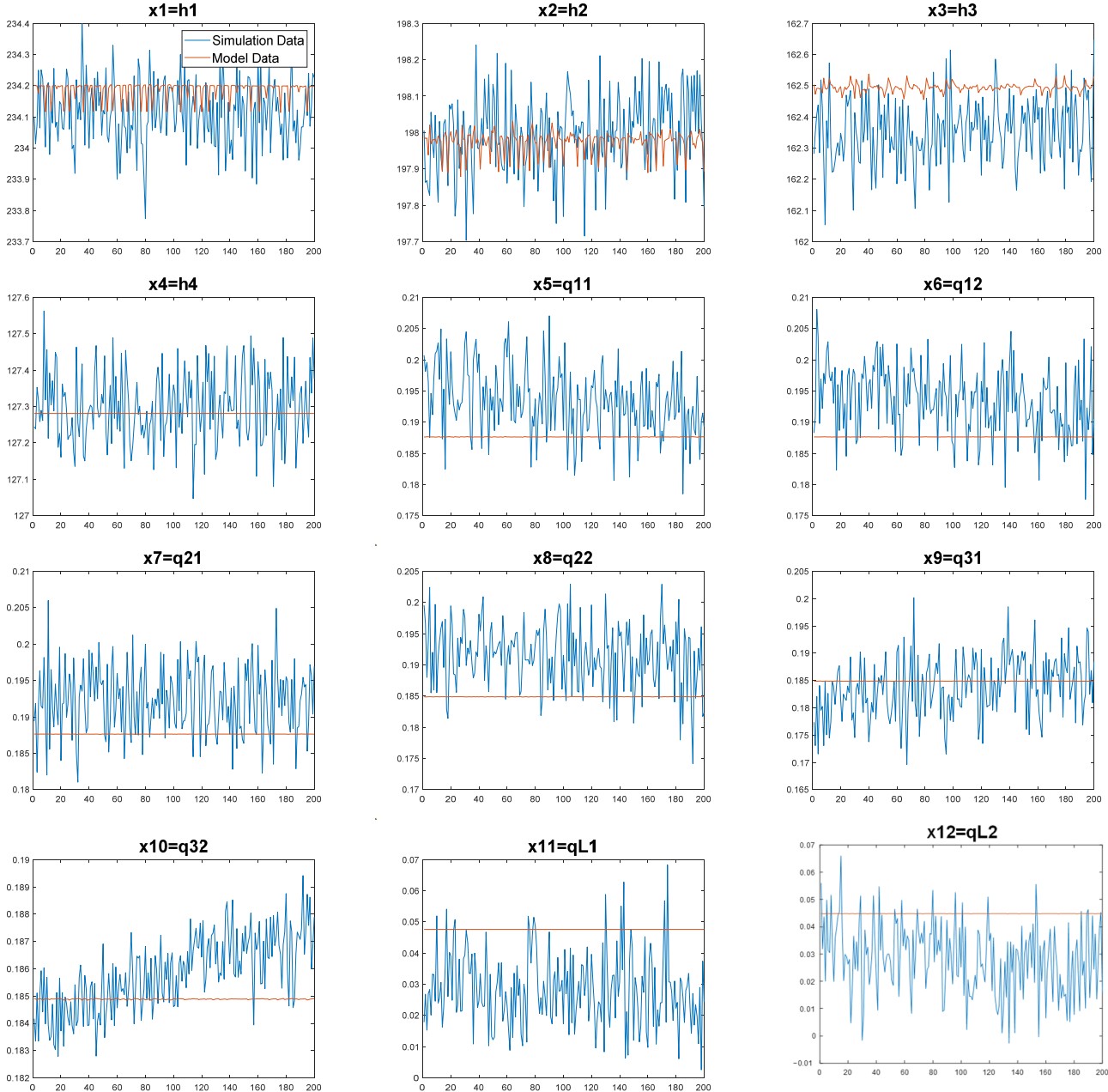

**Figure 8.** Comparison between model data and simulation data hydraulic pressures and flow rates for leakage at 1400 m.

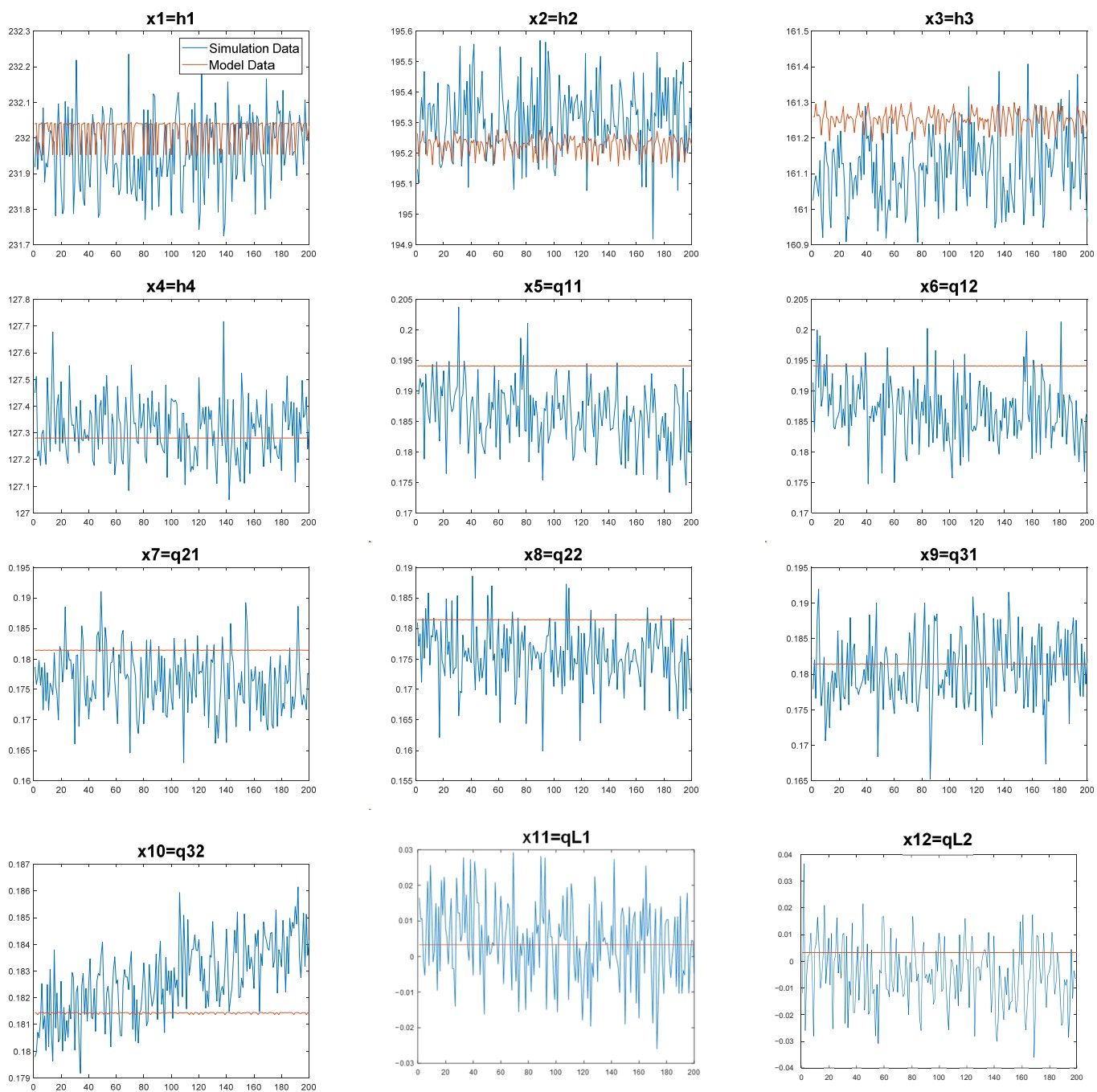

**Figure 9.** Comparison between model data and simulation data hydraulic pressures and flow rates for leakage at 2880 m.

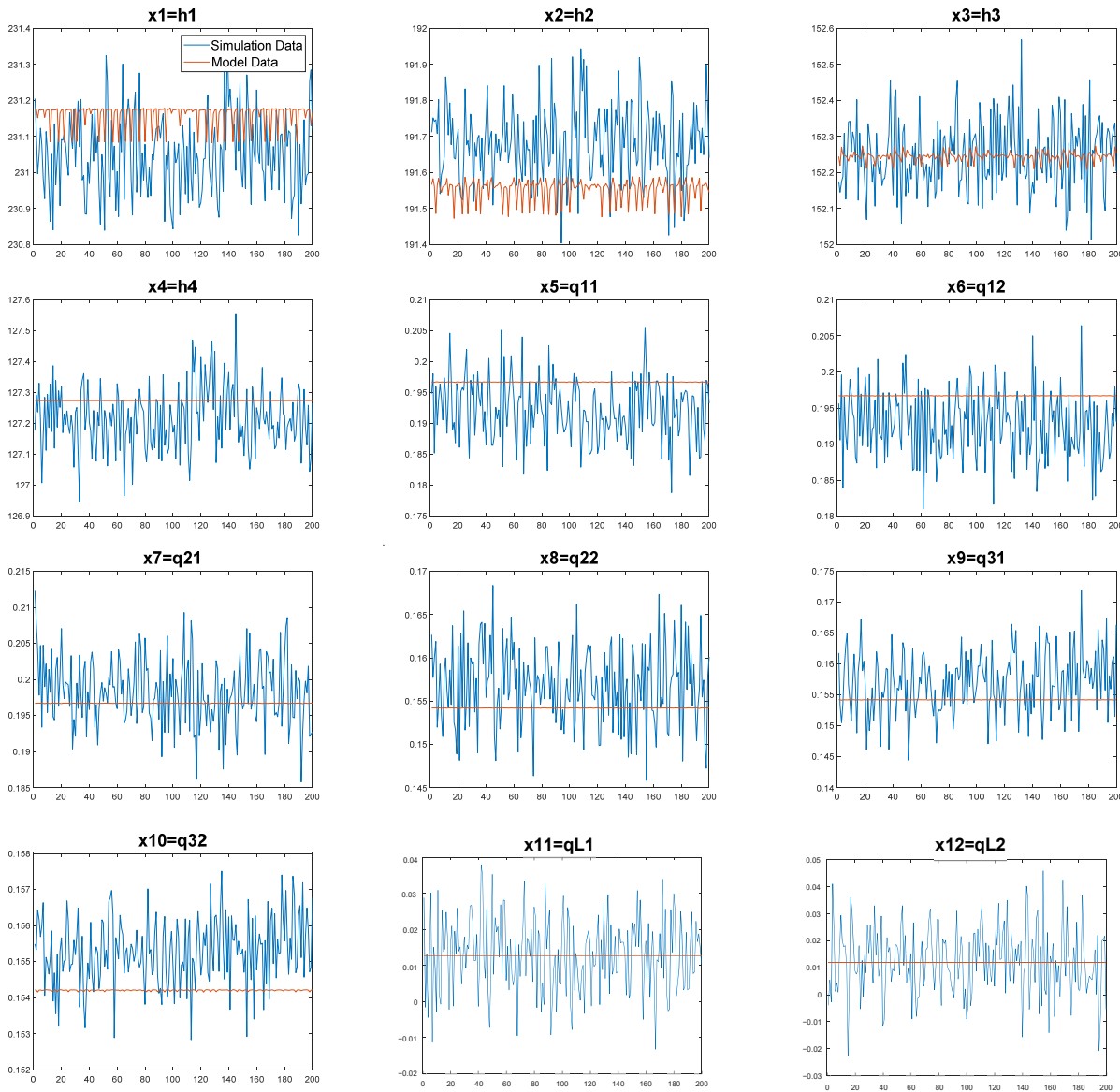

**Figure 10.** Comparison between model data and simulation data hydraulic pressures and flow rates for leakage at 4500 m.

### 4.5. Evaluation of RTTM vs. MRTTM

A comparative analysis between RTTM and MRTTM methods was conducted, with results shown in Table 5 and Figure 11. MRTTM's superior performance in terms of leak location accuracy is highlighted, demonstrating its effectiveness and reliability in pipeline leak detection. Three leakage forms of varying sizes and locations were tested and compared under the same conditions as the methods described in the preceding sections. The RMSE, MAPE, and PDF methods were used for comparison.

**Table 5.** Comparison between the results of RTTM and MRTTM methods.

| | Methods | | | |
|---|---|---|---|---|
| **Leak Location (m)** | **RTTM RSME** | **RTTM MAPE (%)** | **MRTTM RSME** | **MRTTM MAPE (%)** |
| 1400 m | 14.68 | 1.03 | 3.68 | 0.22 |
| 2880 m | 53.81 | 1.86 | 5.9 | 0.16 |
| 4500 m | 22.96 | 0.51 | 3.45 | 0.07 |

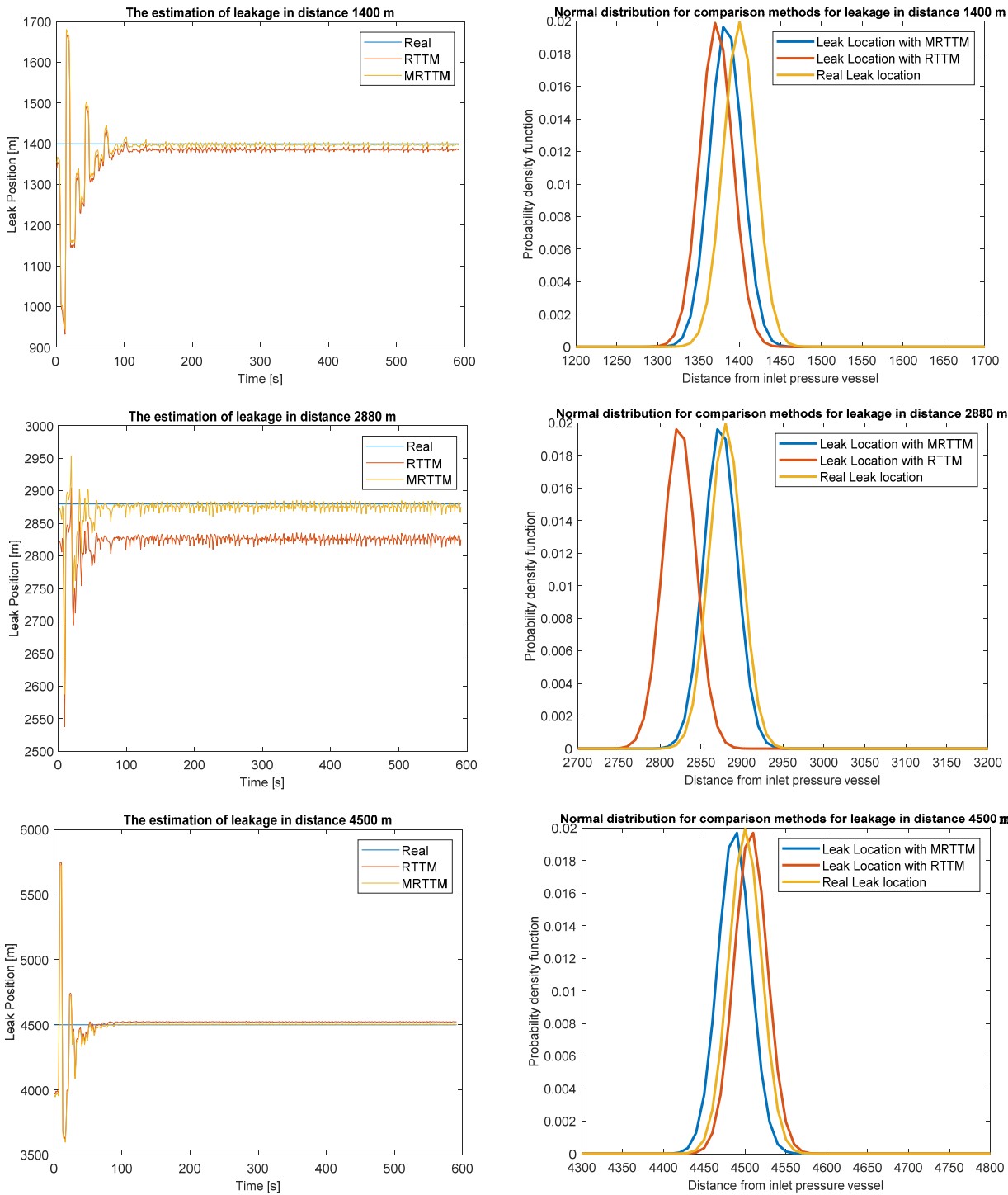

**Figure 11.** Comparison leak detection methods RTTM and MRTTM between the estimation of leakage distance and normal distribution methods for different leakage.

MRTTM significantly outperforms RTTM in leak location accuracy, as evidenced by both Table 5 and Figure 11. Across all three leak scenarios (1400 m, 2880 m, and 4500 m), MRTTM consistently exhibits lower RMSE and MAPE compared to RTTM. This translates to substantially more precise leak location estimates. For example, at 2880 m, MRTTM's RMSE is a mere 5.9 m, representing a nearly 90% improvement over RTTM's 53.81 m. Similarly, MRTTM demonstrates a remarkable reduction in MAPE, decreasing from 0.51% for RTTM to just 0.07% at the 4500 m leak, signifying a significant improvement in the relative error of leak location estimation. This comparative analysis firmly establishes

MRTTM as the superior method for LDS. Its consistent outperformance in both RMSE and MAPE across diverse leak scenarios underscores its effectiveness in providing more accurate and reliable leak location information. This makes MRTTM a highly promising approach for enhancing the overall efficiency and effectiveness of pipeline leak detection and management strategies.

## 5. Conclusions

This study presented the MRTTM framework, a novel approach that leverages AI and advanced signal processing to significantly enhance pipeline leak detection and localization. The MRTTM framework addresses the limitations of traditional methods by incorporating AI for pattern recognition, state space modelling for leak segment identification, and the EKF for precise leak location estimation. The comparative analysis clearly demonstrates the superior performance of MRTTM compared to RTTM, with consistent and significant reductions in both RMSE and MAPE across all leak locations, solidifying its effectiveness as a more accurate and reliable technique. The MRTTM framework is applicable to various types of pipelines, including oil pipelines, gas pipelines, and water transport pipelines. This broad applicability highlights its versatility and robustness in different operational contexts. Additionally, the framework's adaptability to diverse pipeline conditions was briefly discussed, ensuring that its scope of applicability is well-defined. The comprehensive evaluation of all relevant factors prior to decision-making further minimizes the probability of errors and enhances prediction efficiency. Ultimately, the proposed MRTTM framework offers a more accurate, efficient, and reliable solution for pipeline leak detection and localization, significantly reducing the environmental and economic risks associated with pipeline failures. Future research could explore the application of MRTTM to a wider range of pipeline configurations and fluid types, further consolidating its position as a robust and dependable leak detection solution.

**Author Contributions:** Conceptualization, methodology and software, S.A.M.T.; validation, M.M. and S.V.N.; investigation and analysis, M.M. and M.R.S.; writing, S.A.M.T. and S.V.N.; review and editing, M.M. and M.R.S. All authors have read and agreed to the published version of the manuscript.

**Funding:** This research received no external funding.

**Data Availability Statement:** The data presented in this study are available on request.

**Conflicts of Interest:** The authors declare no conflict of interest.

## Nomenclature

| Variables | Description |
| --- | --- |
| $H_{in}$ / $H_{out}$ | Head pressure at the beginning/end of the pipeline |
| $R_{in}$ / $R_{out}$ | The measured difference between real and simulated input/output |
| $h$ | Pipeline's pressure head (m) |
| $q$ | The pipeline's flow rate ($\text{m}^3$/s) |
| $c$ | Fluid's wave speed (m/s) |
| $g$ | The pipeline's gravitational acceleration ($\text{m/s}^2$) |
| $A$ | Pipe's cross-sectional area ($\text{m}^2$) |
| $d$ | Pipe's diameter (m) |
| $f$ | Coefficient of friction |
| $t$ | Time (s) |
| $q_L$ | Leakage stream |
| $h_L$ | Leakage pressure head |
| $\lambda$ | Leakage constant |
| $q_L$ | Leakage flow rate |
| $q_1$ / $q_2$ | Input/output flow rate |
| $Re$ | Reynolds number |
| $X$ | Sensor data collected from the pipeline system |

| | |
|---|---|
| $X_{LPB}$ | Data transferred from the LPB |
| $M_i$ | The $i^{th}$ machine learning model |
| $\hat{L}$ | Estimated leak location |
| $P_I / P_O$ | Pressures measured at the inlet/outlet |
| $\dot{M}_I / \dot{M}_O$ | Mass flow calculated at the inlet/outlet |
| $M_I / M_O$ | Mass flow measured at the inlet/outlet |
| $\dot{M}_{Leak}$ | Leakage rate calculated |
| $X_{Leak}$ | Leak location |
| $T_s$ | Time step |
| $K$ | Kalman's gain |
| $\hat{x}_k \hat{x}_k$ | Expected state |
| $P_k$ | Estimation error covariance matrix |
| $\hat{x}_{k+1}$ | Predicted state |
| $P_{k+1}^-$ | The covariance matrix for predicted error |
| $R$ | Noise measure |
| $Q$ | Processing covariance matrices |
| $x^M$ | Pressure sensor |
| $C^d$ | Leak discharge coefficient |
| $A^{L_n}$ | Leak orifice flow area |

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
