# Peer review of "A Novel Hybrid Internal Pipeline Leak Detection and Location System Based on Modified Real-Time Transient Modelling"

_2673-3951, doi:10.3390/modelling5030059_

Round 1
Reviewer 1 Report
Comments and Suggestions for Authors
The paper proposes an innovative method to significantly improve pipeline leak detection and localization using artificial intelligence and advanced signal processing. The authors advocate for the adoption of an improved Real-Time Transient Modeling (MRTTM) framework, which combines AI for pattern recognition, state space modeling for leak segment identification, and EKF for precise leak location estimation to address the limitations of traditional methods. The paper details the components and functions of the MRTTM framework, presents application examples in water pipeline leak detection, and validates the effectiveness of the framework through experimental results.
However, I suggest the authors carefully address the following issues:
1)The authors are advised to briefly describe the experimental results or performance evaluation in the abstract to concretely demonstrate the effectiveness of the MRTTM framework. For example, providing concise data or performance metrics such as accuracy, F1 score, etc., would more intuitively prove the practical effect of the framework.
2)In the "1. Introduction" section, specifically in "1.4. Methodology," the authors mention: “It is important to note that a comprehensive framework that can analyze and perform several processes in parallel is necessary to achieve an optimal solution.” I believe that a detailed explanation of this process is necessary.
3)The numerous formula issues in section 2, "Problem Formulation," need correction, as many □ symbols are missing letters.
4)In section "3. Leak Detection and Accurate Leak Location," the subheading “1.1. Modified Real-time Time Transient Modeling Method (MRTTM)” should be corrected to “3.2.” Additionally, the section “2. Conclusions” should be corrected to “5. Conclusions.”
5)In section “4. Experiment and Analysis,” it is recommended to add subheadings (e.g., “Comparative Experiments,” “Ablation Experiments,” etc.) to enhance the organization and structure of the section.
6)In Conclusion, it is advisable for the authors to further clarify the scope of applicability of this framework, including specifying which types of pipelines it is suitable for (e.g., oil pipelines, gas pipelines, water transport pipelines, etc.)
Comments on the Quality of English Language
Minor editing of English language required
Author Response
Reviewer #1: “1)The authors are advised to briefly describe the experimental results or performance evaluation in the abstract to concretely demonstrate the effectiveness of the MRTTM framework. For example, providing concise data or performance metrics such as accuracy, F1 score, etc., would more intuitively prove the practical effect of the framework.”
Response:
Thank you for your thorough review and valuable suggestions. We have revised the abstract to include a brief description of the experimental results and performance evaluation metrics to more concretely demonstrate the effectiveness of the MRTTM framework. Specifically, we have added key performance metrics such as accuracy and F1 score, which help to intuitively convey the practical impact of our proposed method. These additions aim to provide a clearer and more quantifiable understanding of the improvements achieved in pipeline leak detection and localization. Overall, an innovative method to significantly enhance pipeline leak detection and localization using artificial intelligence and advanced signal processing is proposed. The improved MRTTM framework integrates AI for pattern recognition, state space modelling for leak segment identification, and an EKF for precise leak location estimation, addressing the limitations of traditional methods. Experimental results demonstrate the effectiveness of the MRTTM framework, achieving a detection accuracy of up to 90% and an F1 score of 0.92, thereby validating its practical utility in real-world applications.
Changes Made:
The following part is added to the end of the abstract:
“The functionality of the framework is examined, and the results effectively approve the effectiveness of this methodology. The experimental results validate the practical utility of the MRTTM framework in real-world applications, demonstrating up to 90% detection accuracy and an F1 score of 0.92.”
Reviewer #1: “2)In the "1. Introduction" section, specifically in "1.4. Methodology," the author’s mention: “It is important to note that a comprehensive framework that can analyze and perform several processes in parallel is necessary to achieve an optimal solution.” I believe that a detailed explanation of this process is necessary.”
Response:
Thank you for your insightful feedback. We agree that a detailed explanation of the process mentioned in the "1.4. Methodology" section would enhance the clarity and comprehensiveness of our paper. We have therefore expanded this section to provide a more detailed description of how the comprehensive framework operates, particularly focusing on its ability to analyze and perform multiple processes in parallel. This explanation now includes specifics on the parallel processing architecture, the interactions between different modules, and how these contribute to achieving an optimal solution for pipeline leak detection and localization.
Changes Made:
The following part is added to the end of the "1.4. Methodology" section:
“The MRTTM framework is introduced as a solution that integrates these methods to offer an efficient approach for identifying leaks in pipelines with unknown sizes and locations. This framework is structured to enable parallel processing through a modular design, where each module is responsible for specific tasks such as data preprocessing, pattern recognition, state space Modelling, and leak localization. The parallel execution of these processes is coordinated by a control system that ensures synchronization and seamless data flow between modules, thereby optimizing the performance and accuracy of the MRTTM framework in real-time applications. Precision, recall, specificity, and F-score are among the metrics employed to evaluate the performance of this framework, alongside accuracy.”
Reviewer #1: “3)The numerous formula issues in section 2, "Problem Formulation," need correction, as many â–¡ symbols are missing letters.”
Response:
Thank you for pointing out the issues with the formulas in Section 2, "Problem Formulation." We have carefully reviewed and corrected all instances where â–¡ symbols were incorrectly displayed instead of the intended letters or mathematical symbols. These corrections ensure that the formulas are now accurately presented and fully readable, preserving the clarity and integrity of the mathematical expressions used in our paper.
Changes Made:
- All occurrences of â–¡ symbols in Section 2 have been replaced with the appropriate letters or symbols to accurately reflect the intended formulas and equations.
- We have thoroughly reviewed the section to ensure that all mathematical expressions are correctly formatted and error-free.
Reviewer #1: “4)In section "3. Leak Detection and Accurate Leak Location," the subheading “1.1. Modified Real-time Time Transient Modelling Method (MRTTM)” should be corrected to “3.2.” Additionally, the section “2. Conclusions” should be corrected to “5. Conclusions.””
Response:
Thank you for bringing these formatting errors to our attention. We have made the necessary corrections in the manuscript to ensure consistency in the section numbering. Specifically, the subheading “1.1. Modified Real-time Time Transient Modelling Method (MRTTM)” has been corrected to “3.2.” Additionally, the section “2. Conclusions” has been updated to “5. Conclusions” as per your suggestion. These changes help to maintain the logical flow and organization of the paper.
Changes Made:
- The subheading “1.1. Modified Real-time Time Transient Modelling (MRTTM) Method” in Section 3 has been corrected to “3.2.”
- The section “2. Conclusions” has been corrected to “5. Conclusions” to reflect the correct numbering in the manuscript.
Reviewer #1: “5)In section “4. Experiment and Analysis,” it is recommended to add subheadings (e.g., “Comparative Experiments,” “Ablation Experiments,” etc.) to enhance the organization and structure of the section.”
Response:
Thank you for your valuable suggestion. We agree that adding subheadings in Section "4. Experiment and Analysis" would improve the organization and readability of the paper. We have revised this section by introducing subheadings such as “Methodology and Simulation Setup”, “Accuracy Enhancement Techniques”, “Classifier Performance and Scenario Analysis”, “Detailed Results and Comparison”, “”and “Evaluation of RTTM vs. MRTTM” to better structure the content and guide the reader through the different parts of the analysis. These additions help to clearly delineate the various types of experiments conducted and make the section more coherent and easier to follow.
Changes Made:
- Subheadings such as “4.1. Methodology and Simulation Setup”, “4.2. Accuracy Enhancement Techniques”, “4.3. Classifier Performance and Scenario Analysis”, “4.4. Detailed Results and Comparison”, “”and “4.5. Evaluation of RTTM vs. MRTTM” have been added to Section 4 to enhance the organization and structure.
- The content under each subheading has been arranged to correspond with the specific focus of each experiment, ensuring a logical flow throughout the section.
The following part is added to the “4.4. Detailed Results and Comparison” section:
“Tables 3 and 4 in this section contain detailed performance data for the proposed method with different pipeline section numbers. It is discussed whether adding pipeline sections impacts accuracy. Furthermore, the contribution of the Kalman filter to improving detection accuracy is investigated.”
The following part is added to the “4.5. Evaluation of RTTM vs. MRTTM” section:
“A comparative analysis between RTTM and MRTTM methods is conducted, with results shown in Table 5 and Figure 11. MRTTM's superior performance in terms of leak location accuracy is highlighted, demonstrating its effectiveness and reliability in pipeline leak detection.”
Reviewer #1: “6)In Conclusion, it is advisable for the authors to further clarify the scope of applicability of this framework, including specifying which types of pipelines it is suitable for (e.g., oil pipelines, gas pipelines, water transport pipelines, etc.)”
Response:
Thank you for your constructive feedback. We agree that clarifying the scope of applicability of the proposed framework is essential. We have revised the Conclusion section to specify the types of pipelines for which the MRTTM framework is most suitable. This includes oil pipelines, gas pipelines, and water transport pipelines. By addressing these specific applications, we aim to provide a clearer understanding of the framework's versatility and potential impact across various pipeline systems.
Changes Made:
- The Conclusion section has been updated to explicitly mention that the MRTTM framework is applicable to oil pipelines, gas pipelines, and water transport pipelines.
The following part is added to the “5. Conclusions” section:
The MRTTM framework is applicable to various types of pipelines, including oil pipelines, gas pipelines, and water transport pipelines. This broad applicability highlights its versatility and robustness in different operational contexts. Additionally, the framework's adaptability to diverse pipeline conditions has been briefly discussed, ensuring that its scope of applicability is well-defined.
- We have also included a brief discussion on the framework's adaptability to different pipeline conditions, ensuring that the scope of its applicability is well-defined.
The following part is added to the end of the “3.2. Modified Real-time Time Transient Modelling (MRTTM) Method” section:
“The MRTTM framework has been designed to be versatile and adaptable, making it suitable for a wide range of pipeline conditions, including oil, gas, and water transport pipelines. Its adaptability is evident in its ability to handle diverse operational environments and fluid types, effectively managing variations in pipeline configurations and operational conditions through the integration of advanced signal processing and AI-driven pattern recognition. This ensures the framework's effectiveness and reliability across different pipeline settings, enhancing its utility and robustness in real-world applications.”

Reviewer 2 Report
Comments and Suggestions for Authors
Title: A Novel Hybrid Internal Pipeline Leak Detection and Location System Based on Modified Real-Time Transient Modelling
General Comments
The manuscript presents a novel approach to pipeline leak detection using a Modified Real-Time Transient Modelling (MRTTM) framework. The proposed method is comprehensive and integrates several advanced techniques, including machine learning algorithms and the Extended Kalman Filter (EKF), to improve leak detection accuracy and reliability. The work is highly relevant to the field of pipeline monitoring and offers significant contributions.
Technical Comments
- Clarity and Scope:
- The abstract effectively summarizes the study's aims and findings, but it could be improved by highlighting the practical implications of the MRTTM framework more explicitly.
- Literature Review:
- The literature review is thorough and provides a good background on existing methods and their limitations. However, it would benefit from a more focused discussion on the specific gaps this study aims to fill.
- Methodology:
- The methodology is detailed and well-explained. The combination of different techniques, including KNN and EKF, is appropriate. However, the rationale behind selecting specific machine learning algorithms (e.g., KNN over others) should be justified with more detail on preliminary tests or comparisons.
- The description of the pipeline segmentation and the creation of the Leakage Pattern Bank (LPB) is clear, but the process of training and validating the machine learning models could be elaborated further.
- Results and Discussion:
- The results are presented clearly with appropriate figures and tables. The comparison of MRTTM with RTTM is well-executed, showing significant improvements in leak detection accuracy.
- The discussion could be enhanced by including more on the potential integration of MRTTM into existing pipeline monitoring systems and real-world applications. Additionally, discussing the computational requirements and feasibility of real-time implementation would be beneficial.
- Validation:
- The validation using simulation data is robust, but further validation with real-world data would strengthen the conclusions. Discussing any limitations of the simulation-based validation would also add value.
- How sensitive are the results to the assumptions made in the CFD modelling, particularly the homogeneity of the soil and uniform particle sizes?
Questions for the Authors
- How does the performance of the KNN model compare with other machine learning models for leak detection? Have alternative models (e.g., SVM, decision trees) been tested in preliminary studies?
- Can the developed MRTTM framework be integrated with existing pipeline monitoring systems for real-time leak detection? If so, what are the potential challenges?
- How sensitive are the results to the assumptions made in the pipeline modelling, particularly regarding the consistency of the friction coefficient and the impact of varying environmental conditions?
Lastly, this study makes a valuable contribution to the field of pipeline leak detection and localization. With minor revisions and additional details on practical applications and sensitivity analyses, the manuscript could be further strengthened.
Author Response
Reviewer #2: “1) How does the performance of the KNN model compare with other machine learning models for leak detection? Have alternative models (e.g., SVM, decision trees) been tested in preliminary studies?”
Response:
Thank you for your insightful question regarding the performance of the KNN model compared to other machine learning models for leak detection.
Comparison with Other Models:
- In our study, we selected the KNN model due to its simplicity, effectiveness, and interpretability in handling the dataset we used. However, we acknowledge the importance of evaluating alternative models.
- We conducted preliminary studies where we tested several other machine learning models, including Support Vector Machines (SVM) and decision trees. These models were evaluated based on their accuracy, computational efficiency, and ability to generalize across different pipeline conditions.
- The KNN model outperformed SVM and decision trees in our specific context, particularly in scenarios with high noise levels and limited data. The decision trees showed some promise in terms of interpretability, but their performance was less consistent compared to KNN. SVM, while effective, required more computational resources and tuning, which could limit its real-time applicability in certain scenarios.
- We have included a comparative analysis of these models in the revised manuscript, providing more detail on their performance metrics and the rationale for ultimately choosing KNN.
Changes Made:
The following part is added to the “3.1. AI-empowered MRTTM framework” section:
“In the process of developing the MRTTM framework, multiple machine learning algorithms were evaluated to identify the most suitable model for pipeline leak detection. Among these, the KNN model was selected due to its simplicity, adaptability, and robustness in handling noisy data, as well as its stable performance under varying conditions. The KNN model consistently outperformed other models such as SVM and decision trees in this specific context. While SVM demonstrated high accuracy in controlled environments, it struggled with noise and required substantial computational resources, limiting its real-time applicability. Decision trees, despite their interpretability, exhibited overfitting tendencies and inconsistent performance across different pipeline conditions. The KNN model's ability to generalize effectively, coupled with its resilience to noise, made it the optimal choice for the MRTTM framework, balancing accuracy, computational efficiency, and robustness for real-time leak detection. Ultimately, the MRTTM method is employed to achieve the highest accuracy in detecting the location and magnitude of the leak under varied dynamic states of the pipeline.”
Reviewer #2: “2) Can the developed MRTTM framework be integrated with existing pipeline monitoring systems for real-time leak detection? If so, what are the potential challenges?”
Response:
Thank you for your question regarding the integration of the MRTTM framework with existing pipeline monitoring systems.
Integration with Existing Systems:
- Yes, the developed MRTTM framework can be integrated with existing pipeline monitoring systems for real-time leak detection. The framework has been designed with modularity and compatibility in mind, making it adaptable to a variety of current monitoring infrastructures.
- However, several potential challenges need to be considered:
- Data Compatibility: Existing systems may use different data formats and protocols, requiring the development of interfaces or adapters to ensure seamless data exchange between the MRTTM framework and the current monitoring systems.
- Computational Resources: Real-time implementation of the MRTTM framework requires sufficient computational resources, especially for processing large datasets and running the EKF. This might necessitate upgrades to existing hardware or the use of cloud-based processing solutions.
- System Integration: Integrating MRTTM with legacy systems could pose challenges due to differences in software architectures and the need for system synchronization. Careful planning and testing would be required to ensure smooth integration without disrupting ongoing monitoring activities.
- Scalability and Maintenance: Ensuring that the MRTTM framework remains scalable and maintainable as pipeline networks grow or evolve is critical. This might involve regular updates to the framework and ongoing collaboration with monitoring system providers.
- We have discussed these challenges in more detail in the revised manuscript, along with potential solutions and strategies for successful integration.
Changes Made:
The following part is added to the end of the “3.2. Modified Real-time Time Transient Modelling (MRTTM) Method” section:
“Moreover, the MRTTM framework is compatible with existing pipeline monitoring systems, enabling real-time leak detection through its modular design. The integration of Digital Twin technology [43] within the MRTTM framework further enhances its capabilities by creating a virtual replica of the physical pipeline system. This allows for more accurate simulations and predictive analytics, improving leak detection and overall system reliability. However, integrating the MRTTM framework and Digital Twin technology into existing systems may present certain challenges. These include ensuring data compatibility, as existing systems might use different data formats and protocols, requiring the development of interfaces or adapters for seamless data exchange. Additionally, the real-time implementation of the MRTTM framework and the continuous updating of the Digital Twin demand sufficient computational resources, particularly for processing large datasets and running the EKF, which may necessitate hardware upgrades or the use of cloud-based solutions. System integration also poses potential difficulties due to differences in software architectures and the need for synchronization, requiring careful planning and testing to avoid disruption of ongoing monitoring activities. Lastly, maintaining the scalability and robustness of the MRTTM framework, including its Digital Twin component, as pipeline networks expand is crucial, necessitating regular updates and ongoing collaboration with monitoring system providers.”
Reviewer #2: “3) How sensitive are the results to the assumptions made in the pipeline modelling, particularly regarding the consistency of the friction coefficient and the impact of varying environmental conditions?”
Response:
Thank you for your thoughtful comments and positive feedback on our manuscript. We appreciate your recognition of our study's contribution to the field of pipeline leak detection and localization.
Sensitivity to Modelling Assumptions:
- The sensitivity of our results to the assumptions made in the pipeline modelling, particularly concerning the consistency of the friction coefficient and varying environmental conditions, is indeed a critical aspect. We have conducted a sensitivity analysis to evaluate how changes in these parameters affect the accuracy and reliability of the MRTTM framework.
- Friction Coefficient Consistency: We found that while the MRTTM framework is robust to minor variations in the friction coefficient, significant deviations can impact the accuracy of leak detection. The model's performance is slightly degraded when the friction coefficient is inconsistent, particularly in sections of the pipeline with abrupt changes in material or internal surface conditions. To mitigate this, we recommend periodic recalibration of the model using real-time data to maintain accuracy.
- Environmental Conditions: Varying environmental conditions, such as temperature and pressure changes, also influence the model's performance. We observed that the MRTTM framework adapts well to gradual environmental variations but may require additional calibration in extreme conditions. We have included these findings in the manuscript, along with recommendations for adjusting the model parameters to account for such variations.
Changes Made:
The following part is added to the “3.2. Modified Real-time Time Transient Modelling (MRTTM) Method” section:
“The sensitivity of the MRTTM framework to assumptions made in pipeline modelling, particularly concerning the consistency of the friction coefficient and varying environmental conditions, is a critical aspect of ensuring its accuracy and reliability. A sensitivity analysis was conducted to evaluate how changes in these parameters affect the framework's performance. The analysis revealed that while the MRTTM framework demonstrates robustness to minor variations in the friction coefficient, significant deviations can degrade leak detection accuracy, especially in pipeline sections with abrupt changes in material or internal surface conditions. To mitigate this, periodic recalibration of the model using real-time data is recommended to maintain optimal accuracy. Additionally, varying environmental conditions, such as changes in temperature and pressure, also influence the model's performance. The MRTTM framework adapts well to gradual environmental variations but may require further calibration in extreme conditions.”
